# Protocol for the pBDG2 Study: Prospective Evaluation of 1.3-β-D-Glucan in the Peritoneal Fluid for the Diagnosis of Intra-Abdominal Candidiasis in Critically Ill Patients

Emmanuel Novy [1,2,*], François-Xavier Laithier [1], Jeremie Riviere [1], Thomas Remen [3], Marie-Reine Losser [1,4], Philippe Guerci [1,4] and Marie Machouart [2,5]

1 Department of Anesthesiology and Intensive Care Medicine, University Hospital of Nancy, F-54511 Vandœuvre-Lès-Nancy, France; fxlaithier@yahoo.fr (F.-X.L.); j.riviere@chru-nancy.fr (J.R.); mr.losser@chru-nancy.fr (M.-R.L.); p.guerci@chru-nancy.fr (P.G.)
2 Stress Immunity Pathogens Unit, Université de Lorraine, SIMPA, EA 7300, F-54000 Nancy, France; m.machouart@chru-nancy.fr
3 Clinical Research and Innovation Delegation (DRCI), MPI Department, Methodology Unit, Data Management and Statistics UMDS, Nancy University Hospital, F-54500 Vandoeuvre-Lès-Nancy, France; t.remen@chru-nancy.fr
4 Acute and Chronic Cardiovascular Deficiency (DCAC), Université de Lorraine, Inserm, F-54500 Vandoeuvre-Les-Nancy, France
5 Laboratory of Mycology, University Hospital of Nancy, F-54511 Vandœuvre-Lès-Nancy, France
* Correspondence: e.novy@chru-nancy.fr

**Abstract:** Background: The delayed diagnosis of the presence of *Candida* in severe intra-abdominal infections exposes patients to an increased risk of mortality. The prevalence of intra-abdominal candidiasis (IAC) varies with the type of intra-abdominal infection, the underlying conditions and the presence of risk factors for *Candida* infection. This study aims to evaluate the interest of the measure of 1.3-β-D-glucan (BDG) in the peritoneal fluid for the early diagnosis of IAC. Methods and analysis: This is a prospective multicenter (*n* = 5) non-interventional study, focusing on all critically ill patients with an intra-abdominal infection requiring intra-abdominal surgery. The primary objective is to assess the diagnostic performance of the BDG measured in the peritoneal fluid for the early detection of IAC using the *Candida* culture as the gold standard. The secondary objective is to report the prevalence of IAC in the selected population. This study aims to enroll 200 patients within 48 months. By estimating the prevalence of IAC in the selected population at 30%, 50 patients with IAC (cases) are expected. These 50 IAC cases will be matched with 50 non-IAC patients (as a control group). The peritoneal BDG will be measured a posteriori in all of these 100 selected patients. This article presents the protocol and the current status of the study. Only the prevalence of IAC is reported as preliminary result.

**Keywords:** intra-abdominal candidiasis; critically ill; abdominal surgery; 1.3-β-D-Glucan; diagnosis; protocol

## Highlights

- This is the first prospective multicenter trial to investigate the interest of the measure of peritoneal BDG for the early diagnosis of intra-abdominal candidiasis (IAC).
- The known prevalence of IAC in critically ill patients with an intra-abdominal infection requiring abdominal surgery means that 200 patients should be enrolled to obtain at least 50 confirmed IACs.
- Fifty critically ill patients with a confirmed IAC will be compared to 50 patients with non-IAC intra-abdominal infection after matching and the random selection of cases.
- Matching factors include the known confounding factors for the measure of BDG.

## 1. Introduction

Recently, in a narrative review on intra-abdominal infection, the European Society of Intensive Care Medicine highlighted the "peculiar challenges" regarding fungal infections and the "need for more solid evidence to firmly guide the use of rapid fungal diagnostics" [1].

Because of the heterogeneity in patient characteristics, clinical presentations and therapeutic management, intra-abdominal infections are not well defined. Regarding fungal infections in this setting, the term intra-abdominal candidiasis (IAC) is widely used and is defined by sterilely collected peritoneal fluid (PF) cultures that are positive for *Candida* spp., as determined by the signs and symptoms consistent with an active infection [2]. Considered to be the most common type of invasive candidiasis in critically ill patients [3], IAC is associated with mortality rates of approximately 25–60% [4,5]. This high variability in mortality could be explained by the heterogeneity of studying populations [3]. Because of the time needed for the PF yeast culture (3 to 5 days) to develop, a delayed introduction of antifungal treatment may occur [6,7]. Thus, the decision to start an empirical antifungal treatment is based on predictive scores such as the *Candida* score [8], the *Candida* colonization index [9] or the peritonitis score [10]. These scores must be integrated in a global evaluation of risk factors and the patient's underlying conditions. These clinical scores and traditional risk factors of invasive candidiasis were unable to identify patients at risk for IAC [11].

In this context, biomarkers such as serum 1.3-β-D-glucan (BDG) have emerged. This non-culture-based method has primarily been performed in the serum of patients with invasive fungal infections or candidemia [12–15]. However, candidemia is present in only 10–20% of patients with IAC [5,16]. Moreover, up to 17% of candidemic adults had persistently negative BDG during their episode of candidemia [17]. The usage of serum BDG to guide the continuation or interruption of antifungal treatment should be performed twice with a 48-h interval. In the presence of two serum measures <80 pg/mL, the clinician should stop the empirical antifungal therapy [18]. Moreover, as highlighted by experts, to be useful, testing for BDG should be performed on homogeneous high-risk patients with invasive candidiasis, accompanied by specific clinical questions [19].

A retrospective pilot study was conducted, evaluating the diagnostic performance of BDG in the PF in comparison to the peritonitis score, direct examination and peritoneal fungal polymerase chain reaction of the same sample [20]. The study indicates a negative predictive value of 100% with one measure of peritoneal BDG. In a population of secondary peritonitis (*n* = 38), a peritoneal measurement of BDG ≤310 pg/mL could rule out an IAC. Regarding the measurement of BDG in the PF (compared with serum assay, which is the only technique validated by the US Food and Drug Administration), no technical issues were noted. Moreover, this diagnostic strategy was independent of the underlying conditions as well as *Candida* risk factors; it could predict the diagnosis of IAC in comparison with culture-based methods.

**Hypothesis 1.** *We hypothesize that the measure of BDG in the PF could be used for the early detection of IAC. Considering the small sample and the retrospective nature of the pilot study, we aim to confirm these preliminary results in a prospective study by enrolling 200 critically ill patients.*

The present article reports the protocol of the study. No results will be presented, except for the current number of included patients and the prevalence of IAC observed in the first inclusions.

## 2. Materials and Methods

### 2.1. Study Design

This multicenter prospective study is being conducted in five centers in France in tertiary teaching hospitals (Besançon, Dijon, Metz, Nancy and Strasbourg).

The study is classified as non-interventional; thus, the intensive care unit (ICU) management and anti-infective strategies (choice of antibiotics or antifungal treatment) is left to the discretion of the clinicians.

This article was written according to STROBE (strengthening the reporting of observational studies in epidemiology) and STARD (studies of diagnostic accuracy) guidelines.

### 2.2. Patients

The targeted population is all adult patients satisfying the following inclusion criteria (Table 1): (i) covered by health insurance, (ii) admitted in participating ICUs, (iii) presented with intra-abdominal infection as well as *Candida* risk factors (according to French recommendations [21]—these include patients with septic shock, immunosuppressive conditions, organ transplants, inflammatory bowel disease or healthcare-associated peritonitis) and (iv) indicated for abdominal surgery. Intra-abdominal infection, which requires abdominal surgery, are abdominal abscess, peritonitis, purulent or necrotic infection in patients having recent abdominal surgery or intra-abdominal events complicated by gastrointestinal perforation or anastomotic leak.

**Table 1.** Inclusion and non-inclusion criteria. ICU: intensive care unit.

| Inclusion Criteria | Non-Inclusion Criteria |
|---|---|
| Adult Covered by health insurance | Death within the first 24 h of ICU admission Pregnant or lactating woman |
| Admitted in participating ICUs with intra-abdominal infection requiring abdominal surgery | Patient deprived of liberty after administration of juridical decisionPatient under psychiatric care Patient under supervision or legal guardianship |
| After information has been provided and non-opposition has been granted by patient or substitute decision-maker | |

### 2.3. Objectives and Endpoints

The primary objective is to compare the measure of the BDG in the PF in the IAC population (cases) and the non-IAC population (controls).

The secondary objectives are (i) to measure the prevalence of IAC in critically ill patients with *Candida* risk factors and an intra-abdominal infection requiring an abdominal surgery, (ii) to identify the risk factors for IAC, (iii) to assess the diagnostic performance of the BDG measured in the PF for early detection of IAC using *Candida* culture as the gold standard and (iv) to compare peritoneal and serum BDG in the same patient.

### 2.4. Gold Standard Test

An IAC is defined by sterilely collected PF cultures that are positive for *Candida* spp., as determined by the signs and symptoms consistent with an active infection. The *Candida* growth should be performed on mycological media (Sabouraud), at 37 °C. A first result (positive or negative) is given to clinicians after five days. All cultures are stored until 21 days. Species of *Candida* are identified using MALDI-Tof technique.

Accordingly, the case group is composed of critically ill patients who underwent abdominal surgery with pre-operative signs consistent with intra-abdominal infection as well as a positive *Candida* culture. The control group is composed of critically ill patients who underwent abdominal surgery with pre-operative signs consistent with intra-abdominal infection and a negative *Candida* culture.

### 2.5. Enrolment

In each participating center, all consecutive patients who fulfilled the inclusion criteria are being enrolled in the study. The Figure 1 depicts the flow-chart of the protocol. In the five centers currently participating in the pBDG2 study, 70 patients have been included in the trial to date.

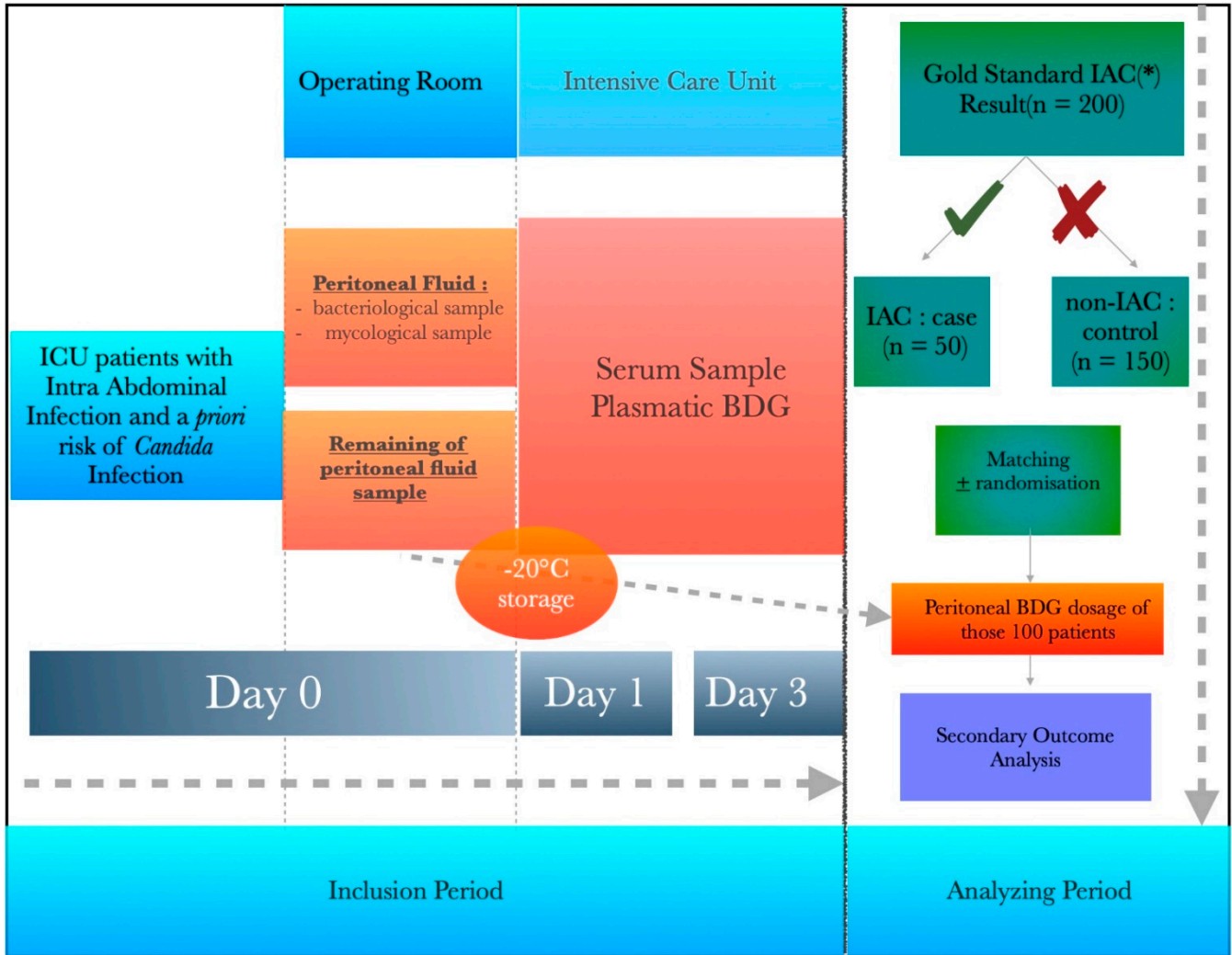

**Figure 1.** Flowchart of the study: from inclusion to analysis. Gold standard IAC: Intra-abdominal Candidiasis (IAC) is defined by sterilely collected peritoneal fluid cultures that are positive for *Candida* spp., as determined by the signs and symptoms consistent with an active infection. BDG: 1.3 Beta D Glucan.

PF samples are being sterilely collected by the surgeon after peritoneal incision with aspiration of 5 mL of peritoneal exudate. This sample is sent to the local microbiological laboratory for routine analysis:

- Laboratory of bacteriology: direct examination (gram stain), culture and antibiotic susceptibility test.
- Laboratory of mycology: direct examination with microscopy, culture and antifungal susceptibility test in cases of non-albicans Candida. Indeed, based on the local ecology of *Candida albicans* resistance in the five centers with no current resistance with fluconazole, antifungal susceptibility test would not be performed in this case.

Then, the remainder of the PF will be stored until the end of the study in a BDG-free device at −20 °C. Fungitell kits (Associates of Cape Cod, Inc., East Falmouth, MA, USA) are already used for serum and will be used for peritoneal BDG measurements. Serum BDG is sampled during the ICU stay on days 1 and 3, according to guidelines [18]. Concerning the measure of the peritoneal 1.3BDG: at the end of the study, if more than 50 patients of the case group are identified, then a random selection will be performed to obtain 50 cases only for further analyses. Each of the 50 remaining patients will be matched with one patient from the control group. The matching criteria are the following: Sepsis-Related Organ Failure Assessment (SOFA) score on the day of the surgery for intra-abdominal infection

(±2), the presence of more than four *Candida* risk factors (parenteral nutrition, pump proton inhibitors, pancreatitis, priori antibiotic exposure <72 h, corticosteroids, renal replacement therapy or *Candida* colonization) and, as a BDG test confounder, the coinfection with documented bacteria. If more than one matching control per case is available, a random selection will be performed to retain one control only per case. All of the 100 peritoneal samples (50 for cases and 50 for controls) will be sent to and analyzed by the laboratory of mycology at the university hospital of Nancy.

### 2.6. Follow-Up

The following baseline characteristics and medical-related variables of patients are being collected from electronic health records during patients' ICU stay: age, sex, body mass index, comorbidities, Knauss and McCabe scores at admission, *Candida* risk factors and surgery data (type of surgery and presence of signs of peritonitis), BDG test confounders [22] (β-lactam exposure, human albumin administration, red blood cell transfusion and renal replacement therapy). The supplementary variables collected are the following: simplified acute physiological score (SAPS II) at admission; SOFA score at admission and at diagnosis of peritonitis; and life-support therapies (vasopressors, renal replacement therapy or invasive mechanical ventilation) as well as their duration of usage and the ICU mortality rate. We are additionally collecting microbiological data, types of antibiotics and types of antifungals used.

### 2.7. Information Provided to Patients

The pBDG2 study is classified as non-interventional. Consequently, the need for informed consent is being waived, and only the "decline to participate" is being requested from patients or substitute decision-makers.

### 2.8. Data Quality, Regulatory Issues and Confidentiality

All personal and medical information is being collected and shared in accordance with medical confidentiality as well as French and European regulations regarding data protection. Furthermore, all data are being recorded in a secured database for statistical analysis with pseudonymization (number of the center, first letter of name and last name). The study data are being collected by each investigator site and are being reported in an electronic case report form. At the end of the trial, the study database and documents will be archived by investigator sites and sponsors in accordance with French and European regulations.

### 2.9. Number of Patients to Include

According to the literature, the prevalence of IAC in severe intra-abdominal infection is estimated to be between 1.5% and 41%. [1] In the pilot study, the rate was 21%. [20] Consequently, the inclusion criteria of severe intra-abdominal infection requiring abdominal surgery and possible *Candida* coinfection indicates an estimate for attempt prevalence of 30%. To obtain at least 50 cases, 165 patients should be enrolled. This number was extended to 200 to facilitate the matching process.

### 2.10. Missing Data

In case of missing data regarding the peritoneal BDG measurements, the baseline and demographics data will be used to study prevalence. If the attempted prevalence in the whole population is less than 30%, then an extension of the number of patients will be requested.

### 2.11. Limitations

For financial reasons, the peritoneal BDG measure could not be performed for all the included patients. Nevertheless, with the inclusion criteria, we hope to obtain a homogenous population of ICU patients with intra-abdominal infection which could limit

the selection bias. Moreover, the process of matching and selection includes a random selection in case of more cases (patients with IAC) than expected.

We fully acknowledged that a recent major surgery is a BDG confounder [23], in particular for its measurement in the serum. Nevertheless, this factor concerns all of the included patients. Moreover, especially in this high-risk post-operative population, the need for a diagnostics biomarker to early identify the presence of fungi is urgently warranted. Lastly, we could hypothesize that the measure of BDG directly in the peritoneal fluid could be less influenced by confounders than its measure in the serum.

### 2.12. Statistical Analysis

#### 2.12.1. For the Primary Endpoint

Student t-tests (parametric) or Mann–Whitney tests (non-parametric) are being utilized to compare BDG concentrations between IAC and non-IAC groups.

#### 2.12.2. For the Secondary Endpoints

The baseline characteristics of the study population are presented as numbers and percentages (qualitative variables including the prevalence of IAC) with the mean and standard deviation or median and extreme values (quantitative variables), depending on the nature of their distribution.

The risk factors of IAC are being explored by employing a univariate and then multivariate logistic regression utilizing a stepwise variable selection (sle = 0.20, sls =0.05).

The threshold of BDG to obtain the optimal specificity and sensitivity is determined by the receiver operating characteristic (ROC) curve.

The concordance between peritoneal and serum BDG levels measured in the same patients is explored through the Bland and Altman method (graphical method). Intraclass correlation coefficients between these variables is also being calculated.

### 3. Trial Status

This trial is registered with clinicalTrial.gov (accessed on 5 September 2019) since 5 September 2019, with the last protocol version 1 dated 3 May 2019. Inclusion started in January 2020 and is expected to end in January 2022. The study was interrupted between March and May 2020 because of the COVID-19 outbreak. Four additional months will be necessary to perform matching, peritoneal sample analysis and statistical evaluation.

In the five centers currently participating in the pBDG2 study, 72 patients have been included in the trial to date. Among these 72 patients, the prevalence of IAC is 47% (*n* = 34/72) with a majority of *Candida Albicans* (60%) species.

### 4. Discussion and Conclusions

This study concerns two areas of research in intra-abdominal infections suggested by international experts in the field:

- The definition and classification of intra-abdominal infection to identify a selected population with high risk of intra-abdominal candidiasis [24].
- As a diagnostic tool, a novel application (peritoneal measurement) of an already existing assay (BDG), which, if proved reliable, could be very useful for early diagnosis of intra-abdominal candidiasis [25].

Indeed, if the peritoneal measure of BDG could confirm or rule out the diagnosis of IAC, then this strategy would allow the rapid introduction of an antifungal treatment or limit excessive exposure. We acknowledge that, currently, routine BDG results are not rapidly available. One of the explanations is the lack of prescription of the BDG test, the cost of the test and the result's delays. Indeed, the real time need for BDG measurement is short. However, because the low number of routine demand and its current cost, most of the analyses are pooled. If we could demonstrate that one test, the day of the surgery, could rule out an IAC and lead to less anti-fungal prescription that are quite expansive, we believe that the routine use of BDG will increase.

**Author Contributions:** Conceptualization: E.N., F.-X.L., T.R., M.M.; methodology, validation: T.R.; data curation: J.R., E.N., F.-X.L.; writing—original draft preparation: J.R., E.N.; writing—review and editing: M.-R.L., M.M.; supervision: P.G. All authors have read and agreed to the published version of the manuscript.

**Funding:** This work was supported by the Department of Anesthesiology, Critical Care and Perioperative Medicine of the University Hospital of Nancy (1000 euros) and a donation from the MSD Laboratory (7000 euros).

**Institutional Review Board Statement:** The study was conducted according to the guidelines of the Declaration of Helsinki and was approved by the ethics committee of Nancy. The study was registered with ClinicalTrials.gov (accessed on 5 September 2019) with the identification number 03997929.

**Informed Consent Statement:** Non-opposition consent is obtained from all subjects involved in the study. In the case of the inability to obtain the patient consent (coma, death before end of participation), non-opposition to participate would be asked to its substitute decision-maker.

**Data Availability Statement:** The current study did not report any data.

**Acknowledgments:** ICU physicians involved in the participating centers: BELLAID Bouhemad (University hospital of Dijon), PILI-FLOURY Sebastien (University hospital of Besançon), POTTECHER Julien (University Hospital of Strasbourg), LOUIS Guillaume (Regional Hospital of Metz-Mercy), Mycologists involved in the participating centers: DALLE Frederic (University hospital of Dijon), GRENOUILLET Frederic (University hospital of Besançon), Valérie BRU (University Hospital of Strasbourg), HOCHARD Hélène (Regional Hospital of Metz-Mercy)

**Conflicts of Interest:** The authors declare no conflict of interest.

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
