# Peer review of "Protocol for the pBDG2 Study: Prospective Evaluation of 1.3-β-D-Glucan in the Peritoneal Fluid for the Diagnosis of Intra-Abdominal Candidiasis in Critically Ill Patients"

_2036-7481, doi:10.3390/microbiolres12010015_

Round 1
Reviewer 1 Report
Authors expose a well done protocol for the use of Beta-glucan measurement to accelerate intra-abdominal candidiasis diagnosis. The study will be really interesting, but the submission did not report any preliminary result neither prevalence of the candida infections in the multicentre study. It's not clear in the introduction, neither in the abstract, that the will of the authors is to describe exclusively the protocol.
Author Response
Dear reviewer,
We are grateful to you for considering a second revision of our manuscript for publication in Microbiology Research.
In the revised manuscript we attempted to address all the issues raised. We would like to thank you for your expertise.
The asked modifications have been inserted in the manuscript, accordingly, with the revision mode.
As you have seen, the article did not report any preliminary result. For this study, classified as non interventional, the protocol didn't plan any analysis after one year of enrollment. Nevertheless, I could report the prevalence of intra-abdominal candidiasis among the 72 first inclusions.
As suggested, in order to clarify the article type (protocol and no original article with results) we had a sentence in the abstract section (P1 - L29,30), in the introduction (P2-L73,74) and as keywords (P1-L32).
Moreover, we add the results concerning the number of included patients (P5-L206) and the prevalence of IAC with species of Candida isolated (P5 - L207,208).
Reviewer 2 Report
This protocol aims to evaluate if peritoneal measure of BDG could confirm or rule out the diagnosis of IAC, which is a life-threatning fungal infection affecting mainly critically ill patients. The objective is important and of great relevance in the field of medical mycology. The study is well structured and the premises as well as the expected results are sound. I have only some minor suggestions:
-It is not clear why antifungal susceptibility testing is planned to be performed only in non-albicans Candida isolates
- Enrollment of patients and ethics statement: there is no mention to an Ethics commitee that would assess if the enrollment of patients, specifically the absence of an informed consent, would be the best approach. Please clarify.
Author Response
Dear reviewer,
We are grateful to you for considering a second revision of our manuscript for publication in Microbiology Research.
In the revised manuscript we attempted to address all the issues raised. We would like to thank you for your expertise.
The asked modifications have been inserted in the manuscript, accordingly, with the revision mode.
Concerning antifungal susceptibility testing, based on the local ecology of Candida albicans resistance in the five centers (and in general in France) with no current resistance with fluconazole, antifungal susceptibility test would not be performed in case of candida albicans. This precision has been added in the manuscript (P3- L121,123)
English language and style have been checked again.
Concerning ethics statement: as requested, we add precision in order to clarify the infromed consent (P6 l247,249). The study was approved by the ethics committee of Nancy (number 234)
Round 2
Reviewer 1 Report
Dear authors, with the clarifications done, as a protocol form, the paper could be accepted